# Analysis of Chlorophyll Concentration in Potato Crop by Coupling Continuous Wavelet Transform and Spectral Variable Optimization

**Ning Liu [1,2], Zizheng Xing [1], Ruomei Zhao [1], Lang Qiao [1], Minzan Li [1,2], Gang Liu [2] and Hong Sun [1,\*]**

[1] Key Laboratory of Modern Precision Agriculture System Integration Research, Ministry of Education, China Agricultural University, Beijing, 100083, China; ningliu@cau.edu.cn (N.L.); S20183081362@cau.edu.cn (Z.X.); S20193081422@cau.edu.cn (R.Z.); b20193080667@cau.edu.cn (L.Q.); limz@cau.edu.cn (M.L.)

[2] Key Laboratory of Agricultural information acquisition technology, Ministry of Agriculture and Rural Affairs, China Agricultural University, Beijing 100083, China; pac@cau.edu.cn

\* Correspondence: sunhong@cau.edu.cn; Tel.: +86-1355-272-6986

**Abstract:** The analysis of chlorophyll concentration based on spectroscopy has great importance for monitoring the growth state and guiding the precision nitrogen management of potato crops in the field. A suitable data processing and modeling method could improve the stability and accuracy of chlorophyll analysis. To develop such a method, we collected the modelling data by conducting field experiments at the tillering, tuber-formation, tuber-bulking, and tuber-maturity stages in 2018. A chlorophyll analysis model was established using the partial least-square (PLS) algorithm based on original reflectance, standard normal variate reflectance, and wavelet features (WFs) under different decomposition scales ($2^1$–$2^{10}$, Scales 1–10), which were optimized by the competitive adaptive reweighted sampling (CARS) algorithm. The performances of various models were compared. The WFs under Scale 3 had the strongest correlation with chlorophyll concentration with a correlation coefficient of -0.82. In the model calibration process, the optimal model was the Scale3-CARS-PLS, which was established based on the sensitive WFs under Scale 3 selected by CARS, with the largest coefficient of determination of calibration set ($R_c^2$) of 0.93 and the smallest $R_c^2 - R_{cv}^2$ value of 0.14. In the model validation process, the Scale3-CARS-PLS model had the largest coefficient of determination of validation set ($R_v^2$) of 0.85 and the smallest root–mean–square error of cross-validation (RMSEV) value of 2.77 mg/L, demonstrating good prediction capability of chlorophyll concentration. Finally, the analysis performance of the Scale3-CARS-PLS model was measured using the testing data collected in 2020; the $R^2$ and RMSE values were 0.69 and 3.36 mg/L, showing excellent applicability. Therefore, the Scale3-CARS-PLS model could be used to analyze chlorophyll concentration. This study indicated the best decomposition scale of continuous wavelet transform and provided an important support method for chlorophyll analysis in the potato crops.

**Keywords:** standard normal variate (SNV); continuous wavelet transform (CWT); wavelet features optimization; competitive adaptive reweighted sampling (CARS); partial least square (PLS)

## 1. Introduction

Potato (*Solanum tuberosum*) is the world's fourth-largest food crop following rice, wheat, and maize [1,2]. Chlorophyll, as the essential photosynthetic pigment of potato leaves, reflects growth

information about plant health [3] and photosynthetic rate [4], and its content is also significantly correlated with the concentration of nitrogen [5]. Therefore, the accurate analysis of the chlorophyll concentration of potato plants is of great importance for nitrogen management in precision agriculture.

Compared with the time-consuming and laborious chemical analysis of chlorophyll concentration [6,7], the modern spectroscopy analysis, as a non-destructive and rapid monitoring method, has advantages in the inversion of chlorophyll concentration of crops due to the principle of light absorption by molecular or chemical bonding [8,9]. At present, the crop analysis method based on spectroscopy primarily includes proximal spectroscopy analysis and remote sensing [10]. The former has advantages of high resolution and accurate data sampling [11]. Thus, it is suitable for the spectroscopy mechanism studies (e.g., the characteristic absorption bands of some material components) and the development of analysis algorithms, thereby laying a foundation for methods of large-scale and large-area remote sensing [12]. Thus, the motivation of this study is to accurately analyze the chlorophyll concentration in potato crops based on proximal spectroscopy.

Three major problems of crop chlorophyll content analysis based on proximal spectroscopy methods include spectral signal-noise reduction, characteristic variable analysis, and analysis model establishment [13,14]. Among them, the noise reduction of spectral data is the primary step to improve the spectral data performance. During spectral data collection, especially in the field environment, noises such as high-frequency noises [15] and scattering effects [16] are inevitably introduced [17]. Accordingly, previous studies have reported that the standard normal variate (SNV) can effectively correct the scattering effect resulting from different light reflection paths to improve the predictive capability of spectral data [18]. The Savitzky–Golay (S-G) smoothing method can reduce the high-frequency noise of spectral data resulting from instrument vibration or electromagnetic interference [19]. However, the main disadvantage of the S-G method is that the smoothing window size is not fixed, which requires complex optimization according to specific spectral data to select the optimal window size [20,21].

Regarding characteristic variable analysis, many methods have been developed to improve accuracy of chlorophyll concentration analysis [22]. One technique is to build a spectral reflectance index; for instance, the normalized difference vegetation index [23], chlorophyll index [24], weighted-difference vegetation index [25], and structural independent pigment index [26] are used to estimate the leaf chlorophyll concentration. Yu [27] reported that the reflectance ratio vegetation index could eliminate the influence of structural difference of wheat canopy on chlorophyll analysis. Another method is to select sensitive wavelengths; for instance, the red edge and characteristic absorption wavelengths are used to analyze chlorophyll concentration. Sun et al. [28] analyzed the spectral migration characteristics of winter wheat at jointing, booting, flowering, and milk-ripening stages. The wavelengths at red edge positions were extracted to establish the chlorophyll analysis models. Five sensitive wavelengths, namely, 680, 716, 1104, 1882, and 1920 nm, were selected to establish the model for chlorophyll and water-content detection.

However, the above methods cannot completely remove noises and present the features of the spectra [29]. Continuous wavelet transformation (CWT) has outstanding quality of time and frequency domain and can decompose a spectrum into numerous wavelet features (WFs) to effectively characterize spectral signals and eliminate high-frequency noises of spectral data [30]. Previous studies [31–33] have reported that continuous wavelet analysis achieves good performance on crop growth-parameter estimation. Li [32] indicated that WFs under $2^3$, $2^4$, and $2^5$ (middle- and low-frequency) scales could reduce the phenomenon of "fingerprint spectrum" with serious vibration noises to improve the analysis accuracy for the leaf nitrogen concentration of wheat and rice crops. The analysis accuracy was higher than the normalized difference vegetation index. Lu et al. [34] indicated that the sensitive WFs of stripe rust and powdery mildew of wheat are distributed in the $2^2$, $2^3$, and $2^4$ scales, and that the WFs could capture the pigment and water content in wheat leaf. These studies show that WFs under middle- and low-frequency scale factors can capture the peak and valley of an absorption feature of physical and chemical materials [33].

In terms of CWT application, some points remain unclear, especially chlorophyll concentration analysis of potato crop during different stages by using spectroscopy combined with the CWT method. Meanwhile, the partial least-square (PLS) regression model is used to explore and evaluate the relationship between scales of WFs and chlorophyll concentration [35,36], which can solve multicollinearity problems [37] among variables by executing a principal component analysis on the independent and dependent variable matrices. Occasionally, the PLS regression model contains uninformative variables, which result in poor prediction accuracy, overfitting phenomenon [38], and further lowering of the model stability. Some studies have reported that in contrast to the correlation analysis [39] and successive projection algorithm [40] methods, the competitive adaptive reweighted sampling (CARS) algorithm, serving as a sensitive wavelength-selection algorithm [41], can improve the performance of the PLS regression model by eliminating invalid variables [42,43]. Thus, we attempt herein to establish a high-performance chlorophyll content analysis model using the CARS-PLS method.

Accordingly, the study aimed to discuss the improvement in spectral data analysis performance by CWT. We focused on the effect of WFs under different decomposition scales on identifying valuable spectral variables and reducing high-frequency noise to enhance the analysis accuracy of potato chlorophyll concentration during growth periods. Combined with SNV correction, we proposed a CWT-CARS-PLS method to establish the high-performance chlorophyll content analysis model. In this model, CWT was used to eliminate the high-frequency noise and extract the valuable spectral variables, and CARS was applied to select WF variables. The effectiveness of CWT in the analysis ability of dynamic estimation of potato chlorophyll concentration was highlighted by comparing and analyzing the model results of original reflectance (Ref) and SNV reflectance (SNV).

The objectives of this study were to (1) determine the dynamic relationships between chlorophyll concentration and canopy spectra at different growth stages; (2) compare the chlorophyll concentration analysis capability of the WFs under different decomposition scales, Ref, and SNV reflectance; (3) use CARS to select sensitive variables and establish various CARS-PLS analysis models; and (4) validate and evaluate the performance of PLS and CARS-PLS models established by Ref, SNV, and different scale WFs.

## 2. Materials and Methods

### 2.1. Data Acquisition

#### 2.1.1. Spectral Data Collection

Experiments were conducted in 2018 and 2020, respectively. Collected data in 2018 were used to develop the spectral variable optimization method and propose the chlorophyll analysis model by CWT. Measured data in 2020 were used to test the proposed method for potato chlorophyll analysis.

In 2018, field experiments were conducted at the National Precision Agriculture Experiment Station in Xiaotangshan, Beijing, China (40°16′25″N, 116°44′03″E). The potato crop was planted on April 10, 2018. According to management practices by farmers, the total N rate was 400 kg N ha⁻¹, with 12% applied at tillering stage, 33% at tuber formation stage, 38% at 127 tuber expansion stage, and the remaining 12% N at tuber maturation stage. Although the growth period of the Atlantic cultivar is about 90 days in Beijing, the canopy changes greatly from the tillering to the tuber maturation stages, after that the leaves of the crop turns to yellow. In order to establish a chlorophyll analysis model to analyze the chlorophyll concentration of potato canopy, as shown in Table 1, spectra data were collected at four growth stages on May 15, May 24, June 7 and June 19, respectively, which were the tillering stage with appearing flower buds (S1), tuber formation stage with flowers (S2), tuber expansion stage after flowers fell (S3), and tuber maturation stage during leaves turning yellow (S4). Eighty plots with a size of 1 m × 1 m were used. Figure 1 shows the location of the field and photos of potato crops for different growth stages. From each growth stage, 80 groups of data were collected, in which 6 groups were invalid because of the influence of low

vegetation coverage. Thus, 74 groups of data were retained at S1. Thus, the modeling dataset had a total of 314 groups of reflectance spectra.

In 2020, the testing experiment was conducted at the Shang Zhuang Experiment Station of China Agricultural University in Beijing, China (40°08′12″N, 116°10′44″E), as shown in Figure 1. The cultivar of the potato crop was Dutch. Due to the epidemic influence in the spring of 2020, the potato crop was planted on 5 June 2020, almost two months later than in 2018, and spectral data were collected on July 11, July 21, July 30 and August 12, respectively. N application and field management practices were similar to experiments in 2018. In addition, the sampling and data collection methods were also the same as in 2018. Although the growth stages of experiments might not exactly match the stages in 2018, collected data could be used to test the effectiveness of proposed methods on the analysis of chlorophyll concentration in potato canopy. Thus, a total of 160 samples were collected from four growth stages as a testing dataset in this paper. Details about the potato growth stages and sampling dates are given in Table 1.

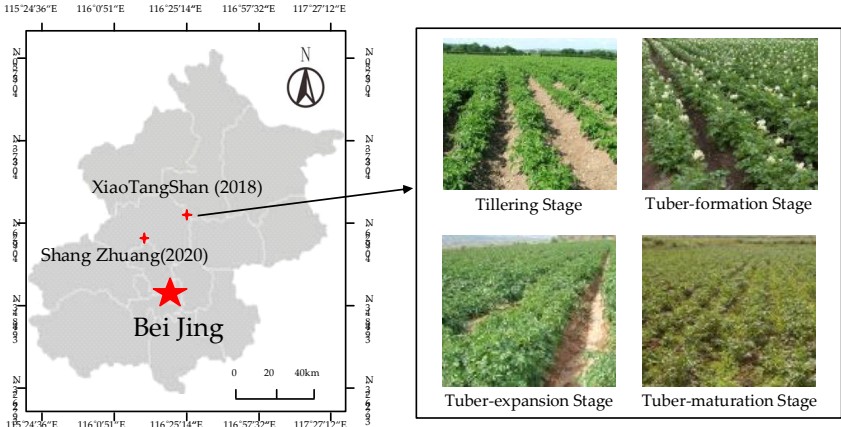

**Figure 1.** Location of the field and photos of potato crops for different growth stages.

**Table 1.** Information of potato crop samples.

| Growth stage | Potato-crop characteristics | Samples | |
|:---:|:---:|:---:|:---:|
| | | Modelling | Testing |
| **S1** | Appearing flower buds, having about 12 leaves | 74 | 40 |
| S2 | Appearing flowers | 80 | 40 |
| S3 | Flowers falling, stems and leaves aging | 80 | 40 |
| S4 | Stems and leaves withering, upper leaves turning yellow | 80 | 40 |

Regarding spectral measurements and leaf sampling, one potato plant was randomly selected in each plot, for which canopy spectral data were collected three times and the average value was calculated to represent the canopy spectrum of the sample. The reflectance spectra were measured by using a ASD FieldSpec-HandHeld-2 spectrometer (Analytical Spectral Devices, Boulder, CO, USA), whose measured wavelength range is 325–1075 nm with step interval of 1 nm, spectral resolution < 3 nm, integration time ≥ 8.5 ms, and standard field-of-view of 25°. There were 751 wavelength variables per spectrum. During data collection, the ASD device was located directly above the sample plant canopy, and the vertical distance from sensor to canopy was about 30 cm. According to geometric operation, the sensor footprint on the potato plant canopy was about 0.02 m². The spectral reflectance was corrected by a standard calibration whiteboard (Spectralon Standard Correction Board, Labsphere Co., Ltd., North Sutton, NH, USA) every 10 min to eliminate the interference of variation in solar-illumination intensity spectral data.

### 2.1.2. Chlorophyll Content Measurement

Three leaves in each sample plant canopy were randomly collected and were put into a freshness protection bag, which was numbered and stored in a portable thermal insulation box. Then, the chlorophyll concentration was determined based on the standard chemical methods in the laboratory [44,59]. Each potato leaf was cut into pieces. About 0.04 g pieces of each leaf were placed in a 25 mL mixture of acetone and anhydrous ethanol to extract chlorophyll. The volume ratio of acetone to anhydrous ethanol was 2:1. The extraction solution was placed in darkness for 24 h. The absorbance at 645 and 663 nm of extraction solution was then measured using a visible-infrared spectrophotometer (UV-752, Shimadzu, Kyoto Japan) that could measure in the wavelength range of 200–1000 nm based on single beam optical system with step interval of 0.1 nm, optical system of a single beam, light source of a tungsten lamp and deuterium lamp, and spectral bandwidth of 4 nm. Chlorophyll concentration was calculated by the following equations:

$$C_a = 12.72A_{663} - 2.59A_{645} \tag{1}$$

$$C_b = 22.88A_{645} - 4.67A_{663} \tag{2}$$

$$C_t = C_a + C_b \tag{3}$$

where $A_{645}$ and $A_{663}$ are the absorbance at 645 and 663 nm, respectively; $C_a$ and $C_b$ are the concentrations of chlorophyll-a and chlorophyll-b, respectively; and $C_t$ is the total chlorophyll concentration, whose unit is mg/L in the study.

### 2.2. Data Analysis

The main data-processing steps are shown in Figure 2. The first part was to convert original reflectance spectra (Ref), which included SNV reflectance (SNV) data obtained from original reflectance by standard normal variate correction, and the wavelet features (WFs) were obtained by continuous wavelet transform (CWT). The second part was to establish analysis models, including PLS models based on the full spectral wavelengths and CARS-PLS models based on the sensitive wavelength variables selected by the CARS algorithm. The third part was to compare the chlorophyll analysis performance of various models.

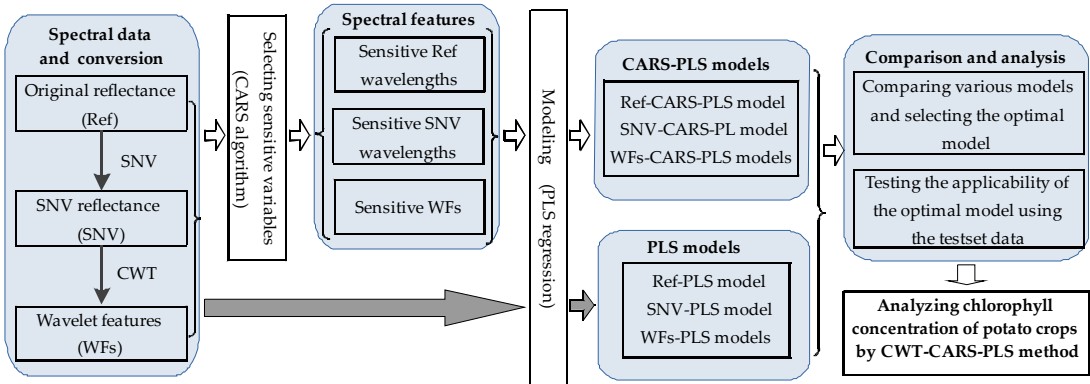

**Figure 2.** Flowchart of the main data processing.

### 2.2.1. SNV Correction

SNV is a certified method that can remove both additive and multiplicative effects in spectral data [45,46]. In SNV, each spectrum was being centered and then scaled by the corresponding standard deviation. It could be calculated with Equation (4):

$$z_i = \frac{x_i - \mu}{\sigma} \tag{4}$$

where $x_i$ is the reflectance of the $i$ nm, $\mu$ is the average reflectance of a spectrum, $\sigma$ is the standard deviation of a spectrum, $z_i$ is the reflectance after SNV of the $i$ nm. In this work, the reflectance spectra corrected by SNV were denoted as SNV reflectance (SNV).

### 2.2.2 CWT

Mathematically, CWT is a liner operation that performs the convolution of reflectance spectrum with a scaled and shifted mother wavelet. The transform process is shown as Equation (5):

$$W_f(a,b) = \frac{1}{\sqrt{a}} \int_{-\infty}^{+\infty} f(\lambda) \bullet \psi(\frac{\lambda-b}{a}) \bullet d\lambda \tag{5}$$

where $\psi(\lambda)$ is the mother wavelet function, $f(\lambda)$ is the reflectance spectrum, and $W_f(a,b)$ is the wavelet coefficient (denoted as $WF_{a,b}$) for the scaling factor $a$ and the shifting factor $b$. The scaling factor indicates the width of the scaled mother wavelet. The scaling factor used in this study was at dyadic scales $2^n (n=1,2,\cdots,10)$, denoted as scale 1, scale 2, …, scale 10, sequentially. The shifting factor was the central wavelength of the shifted mother wavelet. The physical and chemical components of crops had characteristic spectral absorption. $b$ could be used to capture the peak and valley of an absorption feature, and the scaling factor $a$ could be comparable with the width of an absorption feature. A crop leaf reflectance spectrum in the 325–1075 nm range consisted of a background continuum on which a number of absorption features attributable to pigments, water, and dry matter were superimposed [30]. Previous research had suggested that the shape of the absorption features is similar to that of the Gaussian function [47] or a combination of multiple Gaussian functions [48]. Thus, the second derivative of Gaussian, also known as the Mexican Hat, was used as the mother wavelet function in this study. All CWT operations were accomplished using the IDL 6.3 Wavelet Toolkit (ITT Visual Information Solutions, Boulder, CO, USA).

The one-dimensional SNV spectra were transformed into two-dimensional wavelet power map data composed of scaling (frequency scale) and shifting (spectral wavelength) factors by using the CWT. According to previous literature, the scaling factor from 1 to 3 belongs to low frequency, the scaling factor from 4 to 7 belongs to middle frequency, and the scaling factor from 8 to 10 belongs to high frequency [30–34]. The sensitive spectral variables of potato chlorophyll could be selected from these wavelet coefficients.

### 2.2.3 CARS

CARS, proposed by imitating the "survival of the fittest" principle of the Darwinian theory of evolution, is an efficient strategy to select sensitive variables depending on the absolute values of regression coefficients ($|\alpha|$) [43]. The steps of CARS can be summarized as follows [49,50]. First, $|\alpha|$ values are computed and used as indices to evaluate the importance of each variable. Second, the $N$ subsets are selected by $N$ Monte Carlo sampling runs based on the $|\alpha|$ of each variable. Third, a two-step procedure involving an exponentially decreasing function (EDF) and adaptive reweighted sampling (ASR) is used to select sensitive variables. In this step, EDF is utilized to remove the variables whose regression coefficients are relatively small in each sampling run. Following a decrease in EDF-based enforced variables, ARS is used to further eliminate the variables through a competitive way. Finally, the above three steps are repeated until the standard error of cross-validation is obtained, and then the optimal subset of variables is selected.

### 2.2.4. PLS Method

The PLS regression method proposed by Geladi [51] was used to solve multicollinearity problems among variables. PLS regression simultaneously executed principal component decomposition on the spectral reflectance matrix and the leaf chlorophyll concentration matrix [52], which were correlated in the decomposition process. A linear regression model was then established between them to analyze the chlorophyll concentration of potato leaves. To prevent model overfitting, internal interaction verification was performed by leave-one-out cross-validation

(LOOCV), and the optimal latent variation was selected based on the largest coefficient of determination of the cross-validation set ($R^2_{cv}$). The program package of SNV, CARS, and PLS algorithms is available at the http://www.libpls.net/index.php.

### 2.2.5. CWT-CARS-PLS

A new spectral data analysis method named CWT-CARS-PLS was proposed in this study. The sensitive variables selected by CARS can remove the uninformative variables and enhance the PLS model performance. Thus, CARS combined with PLS regression (CARS-PLS) was an effective algorithm to establish the quantitative analysis model. CWT can also transform the one-dimensional SNV spectra into two-dimensional wavelet coefficients. Regarding decomposition, CWT can reduce the high-frequency noises of spectral data and extract the valuable spectral variables. Then, CWT combined with CARS-PLS (CWT-CARS-PLS) can deeply identify sensitive WFs and establish a high-performance analysis model. The proposed CWT-CARS-PLS algorithm is briefly introduced in Figure 3. All data calculations including SNV correction, PLS, CARS-PLS, and CWT-CARS-PLS were completed using MATLAB R2018a software.

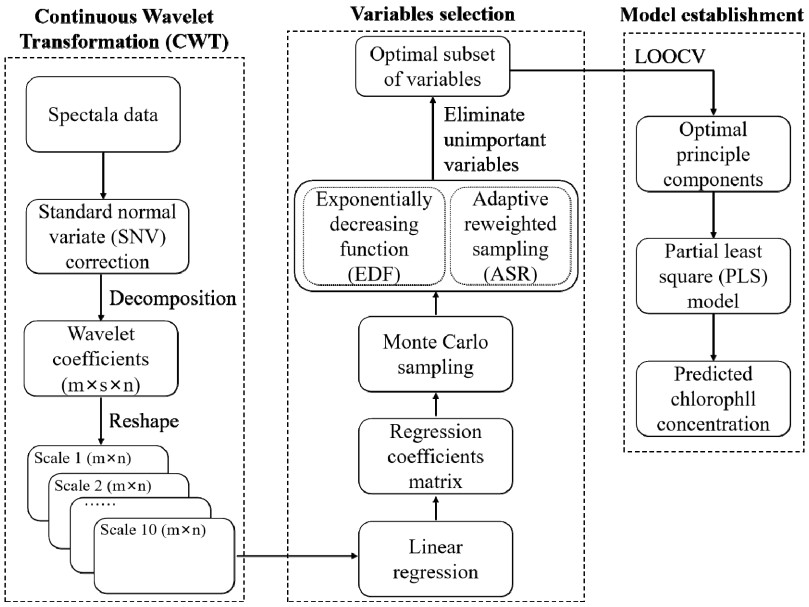

**Figure 3.** Flowchart of Continuous wavelet transformation-competitive adaptive reweighted sampling-partial least-square (CWT-CARS-PLS) algorithm.

### 2.3. Model Evaluation Indicators

To establish the analysis model, the modelling dataset was divided into a calibration and a validation set through sample-set partitioning based on the joint X-Y distance (SPXY) algorithm. This algorithm can comprehensively differentiate independent and dependent variables among samples [53–54].

The calibration set (200 samples) was used to train the PLS model. The validation set (114 samples) was used to verify the established analysis model's performance. The performance of the PLS model was evaluated with the determination coefficient of validation set ($R^2$) and the root–mean–square error (RMSE) as follows:

$$R^2 = 1 - \frac{\sum_{i=1}^{n}(y_i - y_i^*)^2}{\sum_{i=1}^{n}(y_i - \overline{y})^2} \tag{6}$$

$$RMSE = \sqrt{\frac{\sum_{i=1}^{n}(y_i - y_i^*)^2}{n}} \qquad (7)$$

where $y_i$ and $y_i^*$ are the measured and predicted chlorophyll concentrations for sample $i$, respectively. $\bar{y}$ is the average value of measured chlorophyll, and $n$ is the number of samples applied for the calibration or validation set. The difference value ($R_c^2 - R_{cv}^2$) between the $R^2$ of calibration set ($R_c^2$) and $R^2$ of cross-validation ($R_{cv}^2$) can be used as an indicator to judge the model stability, and a smaller value of $R_c^2 - R_{cv}^2$ value implies a more stable model. Furthermore, the $R^2$ of validation set ($R_v^2$) and the *RMSE* of validation set (*RMSEV*) can be utilized to evaluate the PLS model accuracy, and a higher $R_v^2$ and smaller *RMSEV* indicate a better model with stronger predictive capability.

## 3. Results

### 3.1. Statistics on Chlorophyll Concentration of Modeling Data

Chlorophyll concentrations were measured from S1 to S4. The average value at each stage was calculated and used to indicate the dynamic changes of potato growth. Results are shown in Figure 4. Chlorophyll concentration increased from 28.12 mg/L at S1 to 31.04 mg/L with the highest value at S2, and then decreased gradually to 15.36 mg/L at the S4 stage.

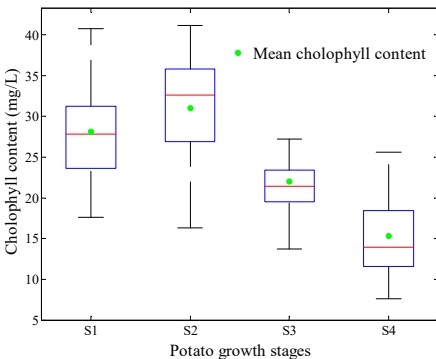

**Figure 4.** Statistical box line graph of chlorophyll concentration during potato growth stage for modeling dataset collected in 2018.

The results of the dataset partitioned by the SPXY algorithm were shown in Table 2, which shows the statistical description of the sample set for each growth stage and the combination of data from all four stages. Samples from all growth stages were combined to represent the changes in chlorophyll concentration. The modelling dataset for the chlorophyll concentration analysis model consisted of calibration and validation sets with 200 and 114 samples, respectively. The maximum value of the calibration set (41.20 mg/L) was larger than that of the validation set (37.46 mg/L), and the minimum value of the calibration set (7.66 mg/L) was smaller than that of the validation set (8.20 mg/L). The division result by SPXY was reasonable, and the calibration set could strongly represent the entire dataset.

**Table 2.** Chlorophyll concentration statistics of modeling dataset.

| Samples | Data set | Sample number | Max (mg/L) | Min (mg/L) | Mean (mg/L) | STD (mg/L) |
|---------|----------|---------------|------------|------------|-------------|------------|
|         | All      | 74            | 40.77      | 17.64      | 28.12       | 5.05       |
| S1      | Calibration | 50         | 40.77      | 17.64      | 28.27       | 5.31       |
|         | Validation | 24          | 33.12      | 19.64      | 27.48       | 3.86       |
| S2      | All      | 80            | 41.20      | 16.30      | 31.04       | 5.81       |
|         | Calibration | 50         | 41.20      | 16.30      | 30.23       | 6.29       |

|  |  |  |  |  |  |  |
|---|---|---|---|---|---|---|
|  | Validation | 30 | 37.46 | 25.26 | 33.45 | 3.04 |
|  | All | 80 | 35.63 | 13.70 | 22.00 | 4.18 |
| S3 | Calibration | 50 | 35.63 | 13.70 | 22.04 | 4.65 |
|  | Validation | 30 | 26.47 | 16.39 | 21.86 | 2.36 |
|  | All | 80 | 32.25 | 7.66 | 15.36 | 5.45 |
| S4 | Calibration | 50 | 32.25 | 7.66 | 15.73 | 5.93 |
|  | Validation | 30 | 20.69 | 8.20 | 14.24 | 3.55 |
| All stages | All | 314 | 41.20 | 7.66 | 24.05 | 7.95 |
|  | Calibration | 200 | 41.20 | 7.66 | 24.07 | 7.95 |
|  | Validation | 114 | 37.46 | 8.20 | 24.00 | 8.00 |

## 3.2. Spectral Data Analysis

### 3.2.1. Analysis of Spectral Response During Growth

Figure 5 (a) shows the Ref curves of the potato crop canopy. Serious scattering effects were observed in the Ref spectra among samples because of the different collection times and light reflection paths. After SNV correction, the noise caused by the scattering effects was significantly eliminated, and the dispersion among spectral curves was significantly reduced, as shown in Figure 5 (b). Accordingly, the SNV spectra were used for subsequent continuous wavelet transformation and modeling analysis.

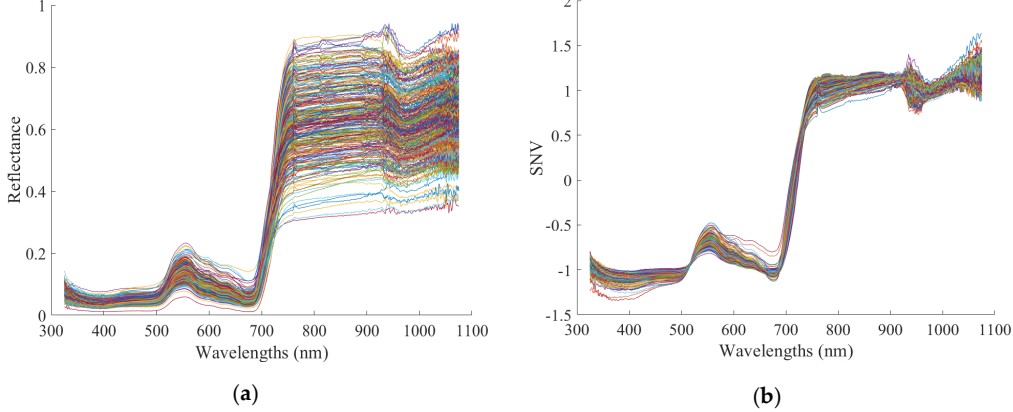

(**a**)  (**b**)

**Figure 5.** Reflectance spectra of potato-crop canopy of four growth stages. (a) Original reflectance (Ref) spectra; (b) standard normal variate (SNV) reflectance spectra.

Furthermore, we examined the dynamic changes between different stages based on the average SNV spectrum of each stage. Figure 6 shows the reflectance of each stage. Their trends were similar in the visible (400–760 nm) and near-infrared (761–1000 nm) regions. In the visible region, the minimum reflectance appeared near 400 and 680 nm due to a strong absorption by the pigment. In the near-infrared region, the reflectance sharply increased from 711 nm to 760 nm because a reflective surface cavity existed in the spongy structure of the mesophyll. Although strong reflection existed in 761–1000 nm as a horizontal platform, a weak reflectance valley appeared near 970 nm because of the weak absorption of leaf water content.

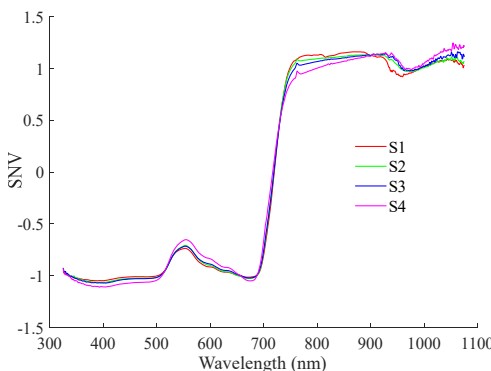

**Figure 6.** Average SNV spectral curve of potato crop canopy per growth stage.

However, significant changes were observed in some specific bands during growth. Within 530–640 nm, the SNV spectral reflectance increased with growth. The average SNV reflectance at S4 was significantly lower than that at the others, whereas the average SNV reflectance of S2 and S3 were very close. Within 740–880 nm, the SNV spectral reflectance decreased gradually. Small reflectance peaks were observed near 763 nm at S2–S4 stages. In the bands of 910–960 nm, the average value at S1 was significantly lower than those at the other stages.

3.2.2. Analysis of Wavelet Coefficient Curves Under Different Decomposition Scales

The SNV spectral curves were decomposed into wavelet coefficients by CWT under 10 decomposition scales. The CWT results for some of the samples are shown in Figure 7. We observed that with increased scale, the wavelet coefficients gradually enlarged and the high-frequency noises were gradually reduced. Thus, the spectral curves were smoothed, and some characteristic absorption peaks were amplified under suitable decomposition scales, as shown in Scales 1–6 (Figure 7). However, when the decomposition scales were too large, the spectral curve became excessively smoothed and caused the the specific characteristic absorption peaks to disappear, which was not conducive to quantitative analysis, as shown in Scales 7–10 (Figure 7).

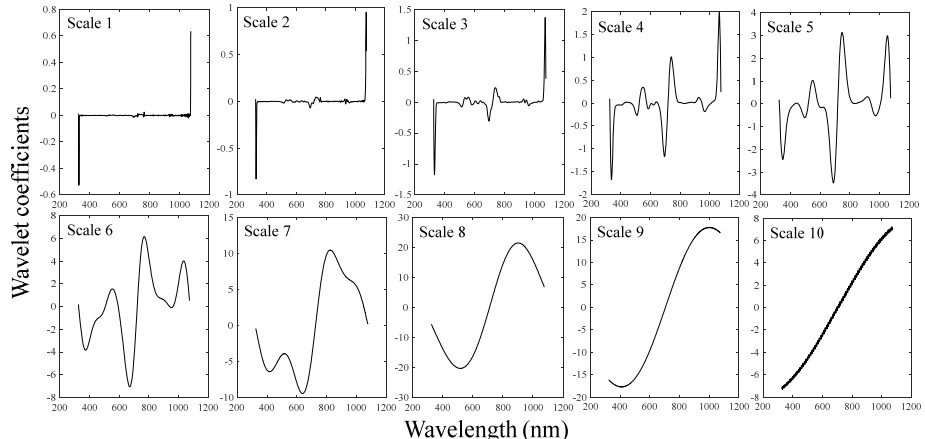

**Figure 7.** Wavelet coefficient curves under different decomposition scales.

*3.3. Correlation of Spectra and Wavelet Features with Chlorophyll Concentration*

3.3.1. Correlation Analysis Between Chlorophyll Concentration and Spectra

Figure 8 shows the correlation coefficient curves between the chlorophyll concentration and Ref and SNV. Compared with Ref, the correlation coefficient between SNV and chlorophyll concentration was higher overall, illustrating that SNV correction reduced the noise of the original

spectra and improved the analysis performance of spectral data. Furthermore, the correlation relationship between the chlorophyll concentration and SNV was analyzed. Within the ranges of 387–509, 519–633, and 744–844 nm, the absolute values of the correlation coefficient ($|r|$) were higher than 0.6. The peak value of the positive correlation occurred at 678 nm, and the $r$ was 0.411. The peak value of negative correlation occurred at 702 nm, and the $r$ was -0.715. Within 845–917 nm, the positive correlation gradually decreased before becoming a negative correlation, and then $|r|$ gradually increased.

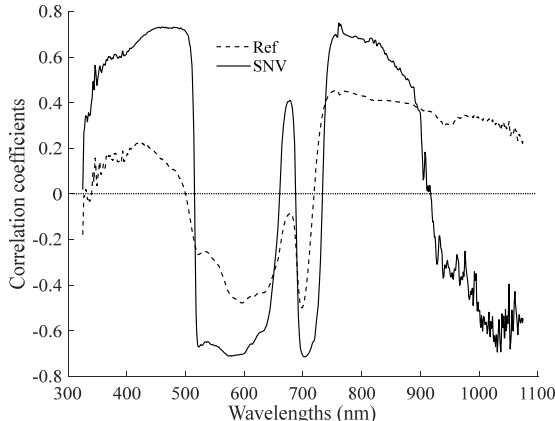

**Figure 8.** Correlation coefficient curve between chlorophyll concentration and spectra.

To further understand how the spectra changed with potato growth, correlation analysis was conducted between SNV and chlorophyll concentration from S1 to S4. Figure 9 shows the correlation coefficient curves. The chlorophyll concentration was correlated positively with the reflectance spectra within the range of 400–500 and 650–700 nm. However, a negative correlation existed between them within 510–630 and 701–750 nm. Furthermore, four band regions were highly correlated, including 400–510, 521–610, 701–740, and 761–920 nm. Overall, the correlation coefficients of S1–S4 had significant differences within 400–600, 601–620, and 700–902 nm. Conversely, the curve trend of the correlation coefficients of S2 and S3 was very similar.

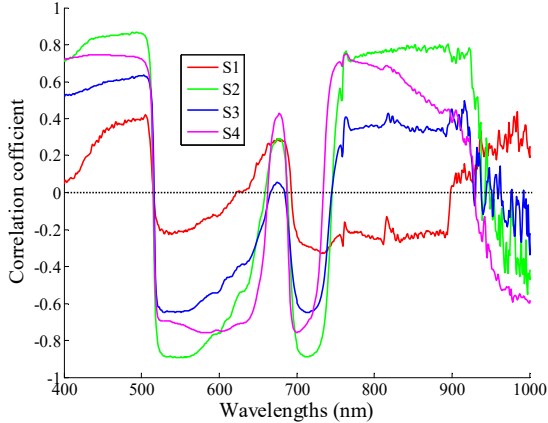

**Figure 9.** Correlation coefficient curve between chlorophyll concentration and SNV.

### 3.3.2. Correlation Analysis between Chlorophyll and Wavelet Features

The correlation coefficients between the chlorophyll and wavelet coefficients were calculated in the decomposition Scales 1–10 to draw the correlation coefficient distribution map, as shown in Figure 10. The correlation coefficient was represented by different colors and color values of each pixel in the map, which could help select the high correlation WFs. We observed that the correlation

coefficients varied in different decomposition scales (scaling factors) and wavelength locations (shifting factors).

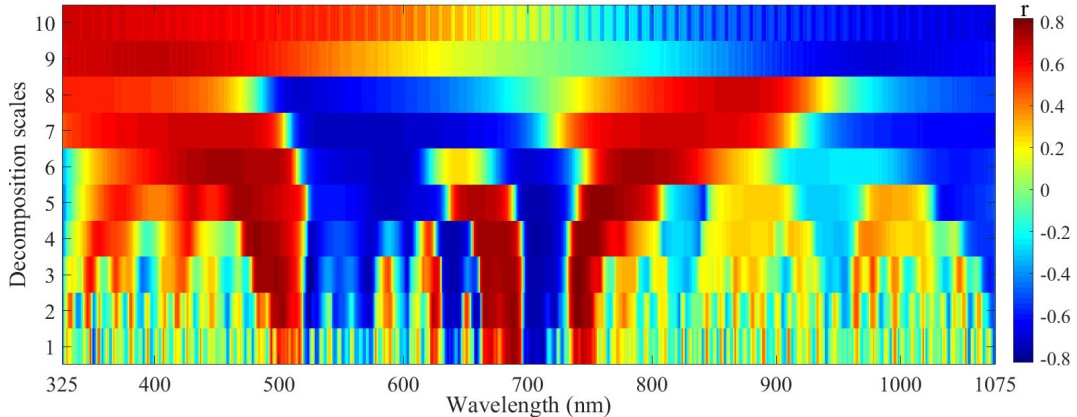

**Figure 10.** Correlation coefficient map between wavelet features and growth stages.

### 3.3.3. Comparison of Correlation Coefficient

The highest correlation coefficients of Ref, SNV, and WFs are shown in Table 3. We observed that the correlation coefficient of SNV (r = 0.75) was higher than Ref (r = 0.50), which revealed that SNV correction could effectively remove the noise of spectral data. For WFs, the correlation coefficient gradually increased form Scale 1 to Scale 3, and then the correlation coefficient gradually decreased. The strongest correlation was found in Scale 3 located in 524 nm (r = -0.82), and the Ref had the weakest correlation (r = -0.50) located in 698 nm.

Moreover, the correlation coefficients of WFs in Scales 1–6 were higher than those of SNV, illustrating that CWT could enhance the correlation of chlorophyll by decomposing spectral data. The correlation coefficients of WFs in Scales 7–10 were also lower than those of SNV, further revealing that spectral data decomposing in too large scales were no longer helpful for quantitative analysis.

**Table 3.** Correlation coefficient (*r*) between chlorophyll and original reflectance, SNV reflectance, and wavelet features.

| | Feature location | Highest *r* |
| --- | --- | --- |
| | Wavelength (nm) | |
| **Ref** | 698 | -0.50 |
| SNV | 761 | 0.75 |
| Scale 1 | 687 | -0.78 |
| Scale 2 | 739 | 0.81 |
| **Scale 3** | **524** | **-0.82** |
| Scale 4 | 744 | 0.78 |
| Scale 5 | 755 | 0.79 |
| Scale 6 | 786 | 0.75 |
| Scale 7 | 547 | -0.74 |
| Scale 8 | 515 | -0.71 |
| Scale 9 | 400 | 0.70 |
| Scale 10 | 1038 | -0.70 |

*3.4. Establishment and Comparison of Chlorophyll Analysis Models*

3.4.1. Sensitive Chlorophyll Variables Selected Using CARS

For the CWT-CARS-PLS, the sensitive WFs in each decomposition scale were selected, and the chlorophyll analysis PLS models were established for every scale. For comparison with CWT-CARS-PLS, the sensitive wavelengths were selected from Ref and SNV data to establish the Ref-CARS-PLS and SNV-CARS-PLS, respectively. The LOOCV was always operated to obtain the optimal principle components (PCs) in establishing the PLS models. The number of variables and PCs of various PLS models are shown in Table 4. For the chlorophyll analysis models, the maximal number of variables was 227 in Scale5-CARS-PLS model, and the minimal number of variables was 31 in Scale1-CARS-PLS. However, the minimal number of PCs was three in Scale3-CARS-PLS.

**Table 4.** Variable information of various CARS-PLS models.

| Models | Variables number | PCs | Models | Variables number | PCs |
|---|---|---|---|---|---|
| Ref-CARS-PLS | 61 | 21 | Scale 5-CARS-PLS | 227 | 21 |
| SNV-CARS-PLS | 64 | 17 | Scale 6-CARS-PLS | 48 | 17 |
| Scale 1-CARS-PLS | 31 | 12 | Scale 7-CARS-PLS | 33 | 15 |
| Scale 2-CARS-PLS | 61 | 13 | Scale 8-CARS-PLS | 54 | 18 |
| **Scale 3-CARS-PLS** | **57** | **3** | Scale 9-CARS-PLS | 57 | 17 |
| Scale 4-CARS-PLS | 178 | 19 | Scale 10-CARS-PLS | 33 | 28 |

The location of sensitive variables selected from Ref, SNV, and WFs in Scales 1–10 by using CARS algorithm are shown in Figure 11. All sensitive wavelengths selected by CARS were distributed in the visible and near-infrared regions. However, for the calibration model established by various sensitive variables, the predictive accuracy of Scale3-CARS-PLS model was the optimum, as shown in Figure 12(a). Furthermore, the sensitive WFs of Scale3-CARS-PLS were analyzed through the leaf information. The number of variables of Scale3-CARS-PLS was 57. These sensitive WFs were located at 346, 389, 419, 425, 426, 431, 435, 436, 437, 520, 523, 535, 546, 547, 563, 579, 580, 590, 591, 620, 625, 661, 662, 667, 684, 685, 688, 690, 693, 698, 716, 717, 733, 739, 742, 751, 752, 767, 781, 811, 824, 825, 848, 857, 858, 875, 890, 909, 919, 929, 939, 948, 960, 963, 968, 973, and 985 nm. Among them, the WFs located in the visible region could reflect the leaf pigment. The WFs located in near-infrared regions could reflect the leaf structure and other leaf substance; for instance, the WF at 929 nm reflected the leaf fat, the WF at 973 nm near 970 nm reflected the leaf water content, and the WF at 985 nm reflected leaf starch.

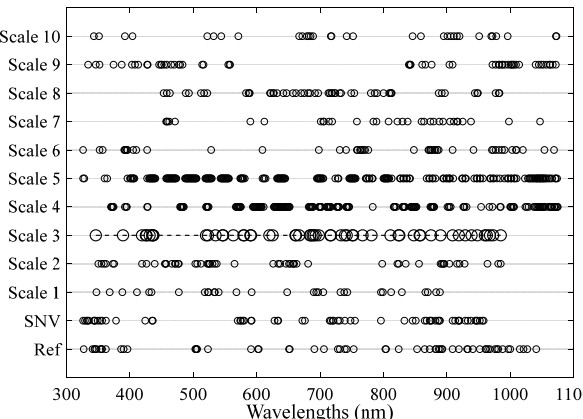

**Figure 11.** Location of sensitive variables selected from Ref, SNV, and wavelet features (WFs) in Scales 1–10 by using the CARS algorithm.

3.4.2. Comparison of the Performance of PLS and CARS-PLS Models

The chlorophyll analysis models were established using the CARS-PLS method, and the modeling results ($R_c^2$ and $R_c^2 - R_{cv}^2$) are shown in Figure 12. To highlight the advantages of

selecting sensitive variables by CARS, the analysis models were also established using the PLS method. Figure 12(a) shows that for all variable categories, the $R_c^2$ of CARS-PLS was higher than that of PLS, illustrating that CARS could effectively eliminate uninformative variables and improve model accuracy. The $R_c^2 - R_{cv}^2$ of CARS-PLS was lower than that of PLS, as shown in Figure 12(b), which revealed that CARS could reduce model complexity and enhance model stability.

Furthermore, the $R_c^2$ of SNV-CARS-PLS was higher than that of Ref-CARS-PLS. The $R_c^2$ of CWT-CARS-PLS models established based on WFs was higher than those of models based on Ref and SNV. For CWT-CARS-PLS, the $R_c^2$ gradually increased from Scale 1 to Scale 3 and then $R_c^2$ gradually decreased. Based on the value of $R_c^2 - R_{cv}^2$, the stability of SNV-CARS-PLS was stronger than that of Ref-CARS-PLS. For CWT-CARS-PLS models, stability gradually strengthened from Scale 1 to Scale 3 and then gradually weakened. The stability of CARS-PLS models based on Scales 1–6 was stronger than those of Ref-CARS-PLS and SNV-CARS-PLS, which was consistent with the correlation analysis in Section 3.3. The above results demonstrated that the CWT could deeply identify spectral data to improve model performance.

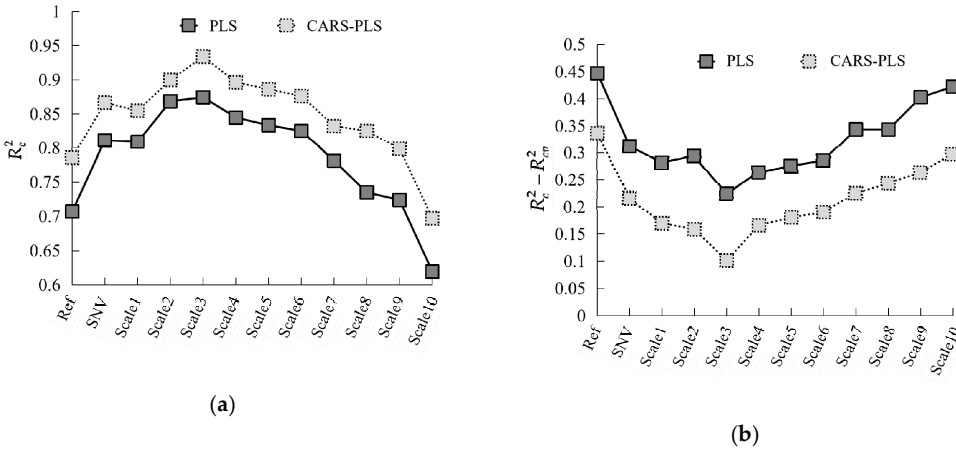

**Figure 12.** Comparison of model performance ($R_c^2$, $R_c^2 - R_{cv}^2$) between PLS and CARS-PLS established by Ref, SNV, and WFs in Scales 1–10. (a) $R_c^2$ of PLS and CARS-PLS; (b) $R_c^2 - R_{cv}^2$ of PLS and CARS-PLS.

### 3.5. Validation of Chlorophyll Analysis Models

The validation results of various chlorophyll analysis models are shown in Figure 13. The same as the calibration models, the CARS-PLS models had a higher determination coefficient of validation set ($R_v^2$) than the PLS models, and the CARS-PLS models had a smaller *RMSEV* than the PLS models primarily because the invalid variables were removed by the CARS algorithm. Furthermore, CWT-CARS-PLS models under Scales 2–6 had higher $R_v^2$ values ($R_v^2 > 0.81$) and smaller RESEV values ($RMSEV < 3.34$ mg/L) than the Ref-CARS-PLS ($R_v^2 = 0.65$, $RMSEV = 4.11$ mg/L) and the SNV-CARS-PLS ($R_v^2 = 0.75$, $RMSEV = 3.55$ mg/L) models. Moreover, Scale3-CARS-PLS showed the highest $R_v^2$ value of 0.85 and the smallest root–mean–square error of cross-validation (RMSEV) value of 2.77 mg/L, as shown in Figure 14. These chlorophyll concentration values were evenly distributed on both sides of the 1:1 line, further illustrating that the proposed Scale3-CARS-PLS model had good stability.

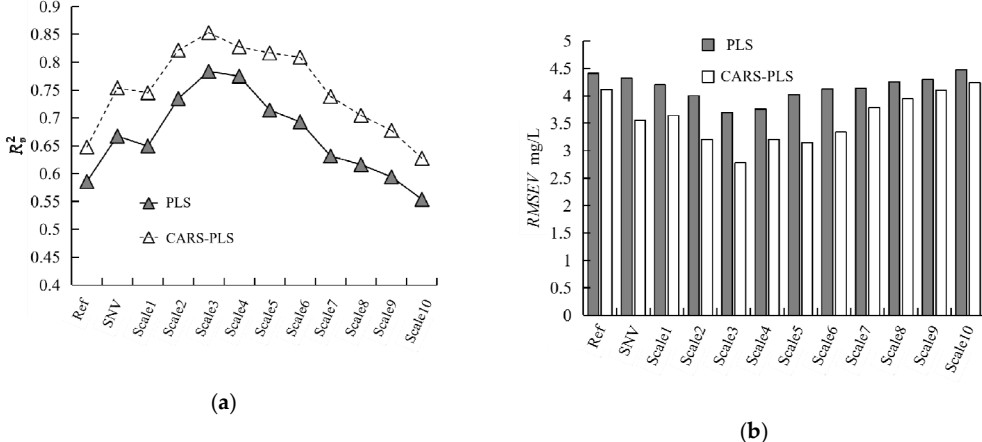

(a)                (b)

**Figure 13.** Validation results of PLS and CARS-PLS established by Ref, SNV, and WFs in Scales 1–10. (a) $R_v^2$ of PLS and CARS-PLS models; (b) root–mean–square error of cross-validation (RMSEV) of PLS and CARS-PLS.

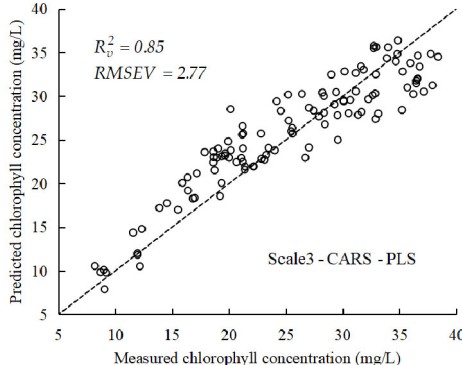

**Figure 14.** Predicted value of validation set for Scale3-CARS-PLS model.

### 3.6. Testing of the Developed Scale3-CARS-PLS Model

The testing set data collected in 2020 were used to test the stability and applicability of the developed Scale3-CARS-PLS model. The chlorophyll concentration of testing set data ranged from 8.81 mg/L to 39.59 mg/L, and the average content was 19.18 mg/L. The chlorophyll concentration range of the test set was smaller than that of the modeling set, which ranged from 7.66 mg/L to 41.20 mg/L.

The canopy reflectance spectra of the testing dataset (160 samples) were corrected by standard normal variate to obtain the SNV reflectance, then the CWT was performed on the SNV reflectance and the CARS algorithm was used to select the sensitive WFs under scale 3, and then the WFs were substituted into the Scale3-CARS-PLS model to predict chlorophyll concentration. In order to highlight the performance of Scale3-CARS-PLS, the reflectance spectra of the testing dataset were substituted into Ref-PLS and Ref-CARS-PLS models, and the SNV reflectance data were substituted into the SNV-CARS-PLS model. The scatter plot of 1:1 was created, as shown in Figure 15, to visually demonstrate the chlorophyll concentration prediction results. The performance of Ref-CARS-PLS was better than Ref-PLS, which showed that CARS could eliminate the valueless variables to improve the model analysis ability. The model performance of SNV-SCAR-PLS was further enhanced due to the SNV pre-processing by correcting the scattering effect. Then, the Scale3-CARS-PLS model showed the strongest $R^2$ of 0.69 and the smallest RMSE value of 3.36 mg/L, which illustrated that the Scale3-CARS-PLS model possessed good analysis capability, and the spectral analysis method had good applicability. Figure 15(d) shows that these chlorophyll values

were evenly distributed on both sides of the 1:1 line, further illustrating that the proposed Scale3-CARS-PLS model had good stability.

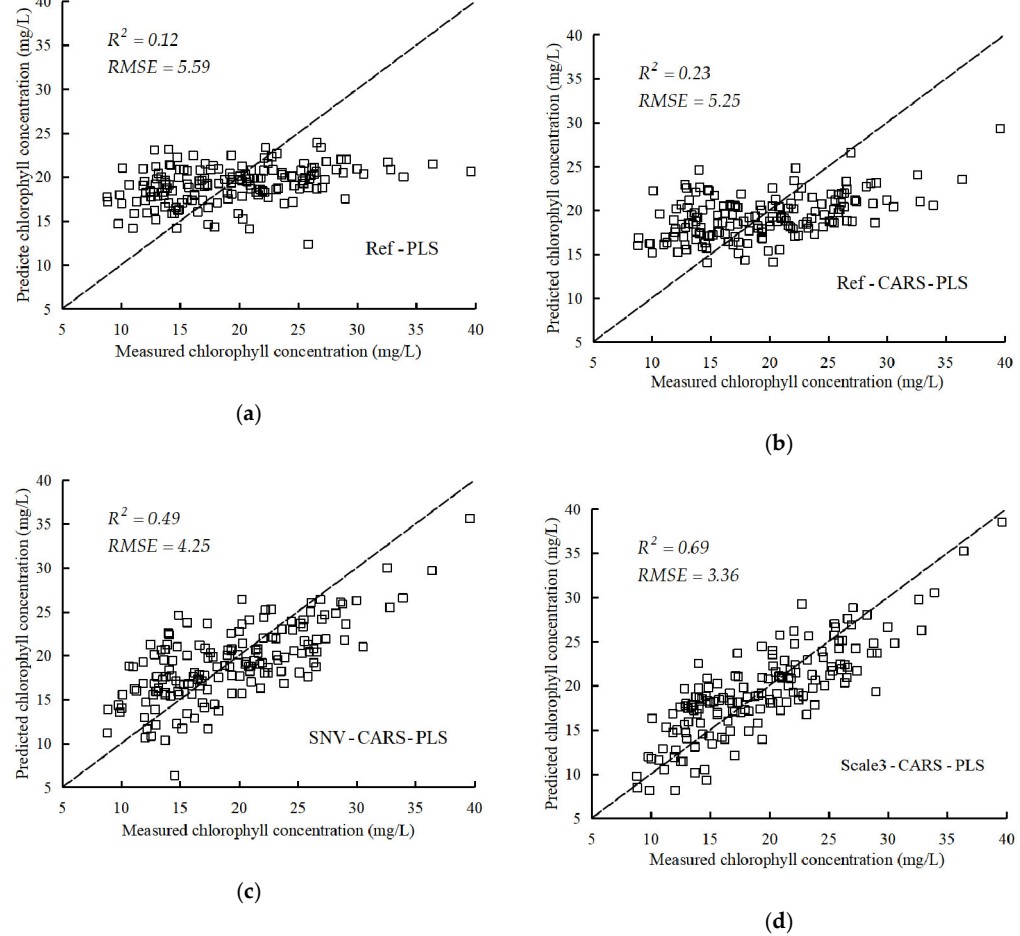

**Figure 15.** Testing results of the various models. (a) Ref-PLS; (b) Ref-CARS-PLS; (c) SNV-CARS-PLS; (d) Scale3-CARS-PLS.

## 4. Discussion

Spectroscopy is a rapid and non-destructive method of gathering crop-pigment information [55,56]. In this study, the spectral characteristic response and chlorophyll concentration change at different stages were analyzed and discussed. Results demonstrated that the average reflectance was close in S2 and S3, and that the correlation curves between the reflectance and chlorophyll concentration of S2 and S3 had similar change trends. According to the potato phenology, a new tuber forms by stolons after the plant flowers at S2 and the tuber expands at S3. Consequently, nutrient availability and balance are transferred from aboveground stems and leaves to underground tubers during these periods. This phenomenon may explain why some of the plants have similar physiology status and spectral responses than others [57].

### 4.1. Abilities of Denoising and Sensitive-Variable Mining of CWT at Different Decomposition Scales

SNV can effectively reduce scattering noise to enhance the analysis performance of spectral data [45,46]. After SNV correction, dispersion among spectral curves was significantly reduced (Figure 5), and the correlation between spectral data and chlorophyll concentration was enhanced (as shown in Figure 8). Accordingly, SNV spectra were used for further CWT and modeling analysis.

After CWT, the spectral reflectance was transformed into the wavelet coefficients, as shown in Figure 7. With increased decomposition scales from 1 to 6, the wavelet coefficient curve was smoothed, and some characteristic absorption peaks amplified. Then, the curve was excessively smoothed, resulting in the disappearance of the characteristic absorption location. The above content was consistent with previous literature reporting that WFs in the middle- and low-frequency scales could capture the absorption characteristics of the physical and chemical substances of crops [33,58] and effectively eliminate the high-frequency noise of spectral data [36,59]. High-frequency WFs could remove the absorption features and could not efficiently analyze the physiological and biochemical compositions [60].

The absolute value of the highest correlation coefficient between chlorophyll concentration and WFs under Scales 1–6 was higher than SNV (0.75), illustrating that the CWT could enhance the correlation of chlorophyll concentration by decomposing spectral data. Previous studies [30,34,59–61] have reported the same results, such as Wang [59] who indicated that the correlation between wavelet coefficients and pigments was significantly higher than that of vegetation index and sensitive wavelengths. Furthermore, with increased decomposition scales from 1 to 3, the absolute value of the highest correlation coefficient of WFs increased from 0.78 to 0.82 and then gradually decreased to 0.70, illustrating that high-frequency WFs were not conducive to quantitative analysis [32,33,60,61]. WFs under Scale 3 exhibited the strongest correlation relationship with chlorophyll concentration.

### 4.2. Uninformative Variable Elimination by CARS Algorithm

Given that a spectrometer collects reflectance data based on near-contiguous spectral bands, the selection of sensitive wavelengths or variables is one of key steps in the chlorophyll analysis to solve multiple mutual lineal problems of overfitting and redundancy [62]. Thus, wavelengths and WFs need be selected by effective algorithms to remove the uninformative variables and to enhance model performance [59,63]. The CARS developed based on the model population analysis strategy [64] can be used to consider the contribution of each variable to the analysis model to select informative spectral variables. Relative to the PLS models, the number of input variables of CARS-PLS models was reduced significantly, and the CARS-PLS models possessed more excellent prediction ability, as shown in Figures 12(a) and 13. Overfitting frequently occurred during the modeling process, caused by the increasing number of model variables, which affected the stability and accuracy of the PLS model [65]. Accordingly, internal cross-validation was performed in this study. The difference in the determination coefficient of calibration and cross-validation sets ( $R_c^2 - R_{cv}^2$ ) was used as an indicator to determine the model stability [66]. As shown in Figure 12(b), the $R_c^2 - R_{cv}^2$ values of the CARS-PLS models were lower than those in the PLS models, further illustrating that the CARS can effectively eliminate redundant variables and improve the analysis of the model's stability.

### 4.3. Chlorophyll Content Analysis Capability of WFs Under Different Decomposition Scales

We further analyzed the performance of various CARS-PLS models. From the point of view of model stability, the $R_c^2 - R_{cv}^2$ values of the CARS-PLS models based on Scales 1–6 were smaller than those of Ref-CARS-PLS and SNV-CARS-PLS, and the $R_c^2 - R_{cv}^2$ value of Scale3-CARS-PLS model was the smallest, showing that Scale3-CARS-PLS model had the strongest stability. From the point of view of prediction capability of the model, the *RMSEV* values of CARS-PLS models based on Scales 2–6 were smaller than those of SNV-CARS-PLS model, and Scale3-CARS-PLS showed the strongest $R_v^2$ value of 0.85 and the smallest RMSEV value of 2.77 mg/L, as shown in Figure 14. For WFs under Scale 3, 57 sensitive WFs were selected by the CARS algorithm, whose locations were evenly distributed in the visible (37 variables) and near-infrared (20 variables) region, as shown in Figure 11. Previous studies have reported that the spectral data in the visible region can analyze pigment content [67]. Moreover, the spectral data in the near-infrared region can reflect other

substances' information and crop-canopy structure, which can improve the robustness of the chlorophyll analysis model [68].

Moreover, previous studies reported the detection of chlorophyll concentration in crops based on spectral wavelengths or/and spectral indices. Sun [28] selected 11 sensitive wavelengths for analyzing the chlorophyll concentration of potato leaf, with the $R_v^2$ of the model of 0.77. Tao [69] screened the red edge position using the linear extrapolation method for estimating the chlorophyll concentration of potato with $R_c^2$ of 0.87. However, the $R_c^2$ and $R_v^2$ of the analysis model developed by coupling CWT with CARS methods in this paper is 0.93 and 0.86, respectively. Above content demonstrated that CWT could deeply identify spectral data to improve model performance, and that the sensitive WFs under Scale 3 possessed the best excellent prediction capability for chlorophyll concentration of potato crops.

### 4.4. Generalizability of This Study to Future Works

A comprehensive analysis of testing results showed that spectral data could be processed using CWT. Sensitive variables were selected using CARS, which was suitable for model-variable optimization and prediction-capability improvement. Finally, the analysis performance of the Scale3-CARS-PLS model was tested using another variety of potato crop, the $R^2$, and RMSE was 0.69 and 3.36 mg/L, as shown in Figure 15, which demonstrated that the Scale3-CARS-PLS model possessed good stability and excellent applicability. Previous studies reported that the chlorophyll concentration is significantly correlated with the concentration of nitrogen [5,70]. Therefore, the study could provide a theoretical support for precision nitrogen management in the potato field, and a method reference for large-scale remote sensing analysis of potato chlorophyll concentration.

However, this method was based on specific spectral data for potato crops. The restrictions were based on the existence of other datasets or potato varieties [22,71]. Therefore, more datasets from wide-ranging potato varieties, planting patterns, and experimental fields should be collected to develop a stable and accurate classification model using CWT-CARS-PLS method.

### 5. Conclusions

We presented an effective method for analyzing the chlorophyll concentration of potato plants through canopy spectroscopy. The dynamic responses of canopy spectra at different growth stages were analyzed. The spectral characteristics were found to significantly differ between S1, S2–S3, and S4. However, the SNV spectral reflectance curves in S2 and S3 were similar. The performances of Ref, SNV, WFs under different decomposition scales, CARS-PLS, and CWT-CARS-PLS in analyzing chlorophyll concentration were compared based on the model results. The CARS-PLS model established by WFs under different scales obtained by CWT exhibited the most excellent analysis ability and reliability. Scale3-CARS-PLS model had fewer variables, smallest $R_c^2$-$R_{cv}^2$ value, strongest $R_v^2$, and weakest *RMSEV* for chlorophyll analysis. The analysis performance of the Scale3-CARS-PLS model was tested using another variety of potato crop with a satisfactory result. Based on spectral data, the WFs under Scale 3 showed excellent chlorophyll-content prediction capability. Thus, the proposed CWT-CARS-PLS was a potentially accurate and efficient method of analyzing the chlorophyll concentration of potato crops. This study could provide a method reference for large-scale remote sensing analysis of chlorophyll concentration and a theoretical support for precision nitrogen management of potato crops.

**Author Contributions:** Conceptualization, Ning Liu, Zizheng Xing, Ruomei Zhao, Lang Qiao, Minzan Li, Gang Liu and Hong Sun; data curation, Ning Liu, Zizheng Xing, Ruomei Zhao, Lang Qiao, Minzan Li, Gang Liu and Hong Sun; formal analysis, Ning Liu, Zizheng Xing, Ruomei Zhao, Lang Qiao, Minzan Li, Gang Liu and Hong Sun; funding acquisition, Minzan Li and Hong Sun; investigation, Ning Liu, Zizheng Xing, Ruomei Zhao, Lang Qiao, Minzan Li, Gang Liu and Hong Sun; methodology, Ning Liu, Zizheng Xing, Ruomei Zhao, Lang Qiao, Minzan Li, Gang Liu and Hong Sun; project administration, Minzan Li, Gang Liu and Hong Sun; Resources, Ning Liu, Lang Qiao, Gang Liu and Hong Sun; software, Ning Liu, Zizheng Xing, Ruomei Zhao and Hong Sun; supervision, Minzan Li, Gang Liu and Hong Sun; validation, Ning Liu, Zizheng Xing and Gang

Liu; visualization, Ning Liu, Zizheng Xing, Ruomei Zhao and Lang Qiao; writing–original draft, Ning Liu; writing–review & editing, Minzan Li, Gang Liu and Hong Sun. All authors have read and agreed to the published version of the manuscript.

**Funding:** The project was supported by the National Natural Science Fund of China (Grant No. 31971785, 31501219), the Fundamental Research Funds for the Central Universities of China (Grant No. 2020TC036), and the Graduate Training Project of China Agricultural University (JG2019004 and YW2020007).

**Acknowledgments:** Thanks to Tao Zheng and Li Wu for their help in the field experiment.

**Conflicts of Interest:** The authors declare no conflict of interest.

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
