# Peer review of "Analysis of Chlorophyll Concentration in Potato Crop by Coupling Continuous Wavelet Transform and Spectral Variable Optimization"

_remotesensing, doi:10.3390/rs12172826_

Round 1

Reviewer 1 Report

The manuscript describes (i) the calibration process of the spectral analysis of leaf chlorophyll and (ii) measurement of a time series during vegetative growth of potato. The approach isn't innovative, but gives small details on the methodology, which ca be interesting to the precision agriculture community.

In the abstract, if an abreviation has been introduced, it should be used such as R². Particularily for R², it represents the coefficient of determination instead of determination coefficient.

The wording "detection" is kind of missleading as it refers to a choice between chlorophyll is / is not present. Later the authors write analysis of chlorophyll "content", which seems to better reflect the meaning in this work.

line 36: check plural

line 37: "ability"?

line 38: remove "significant" as it is already stated that variables are correlated, which wouldn't be the case, if it is not significant.

line 40: avoid "significant", if not referring to a statistical analysis.

It goes on like this with many small needs for revision in almost every line, please revise the language and terminology.

Figure 1 belongs to material and methods.

Chlorophyll analysis: The use of few wavelength for analysing the extract is disappointing considering the background of the authors in spectroscopy. There are whole spectrum methods available, which provide much better reference data. Please see publications of A.R. Wellburn or M. Pflanz.

Data processing: The justifications for the application of certain methods can be given in the introduction and further mentioned when discussing own results. In the material and methods only details on what was done can be given to enable the reader to repeat the study.

Generally, if equipment is named, please provide the measuring principle and type, manufacturer, country in parenthesis behind, such as "... a spectrophotometer (ASD FieldSpec-HandHeld-2, Analytical Spectral devices, USA) was equipped with ....". You may add the city as well before the country.

Provide details on the spectrophotometer such as optical geometry, calibration...

line 225: It is stated "We propose...". Was this done in a former manuscript or in the present one? Please elaborate more on this.

Please find a more suitable heading instead of "Application of data set"

In section 2.3, data are provided, which seem to be results.

line 249: subscript is missing

It seems a cross-validation was carried out. If this is true, please change validation into the more specific term cross validation throughout the text.

In the figures and tables it reads "potato", but the foligae is addressed. Actually the measurements took place on the leaf.

No discussion of results is provided. Please complete the manuscript by comparing and discussing your findings with the work of other researchers.

I wasn't able to check on plagiarism or similarities.

Reviewer 3 Report

Liu and colleagues combined continuous wavelet transformation (CWT), competitive adaptive reweighted sampling (CARS) and partial least squares (PLS) regression to develop the CWT-CARS-PLS approach for estimation potato chlorophyll content. The developed method outperformed the commonly used method and showed potential for improved chlorophyll estimation using hyperspectral reflectance spectra. At this stage, I can only recommend the manuscript for major revision. Please see my detailed comments below.

Line 26, add the unit for RMSE value.

Lines 55-57: needs citations.

Line 83: “… and deeply mine sensitive variable information.” Not sure what do you mean by this.

Show the location of the field in a map and provide filed photos for different growth stages if available.

What’s the sensor footprint on the leaves from the 30cm distance?

How many times each leaf was measured for spectra?

What type of whiteboard used as a normalization standard?

Less than 3 nm is the full-width half maximum of the spectrometer?

Clarify if you resampled to spectra into 1 nm interval for further processing.

What were the criteria for the selection of leaves?

How were the leaves transported to the lab for chlorophyll measurements?

For model validation also add RMSE(%)

Is Fig4 for all growth stages or just for one?

In section 3.4.2, PLS means using original spectral reflectance as input? What about CARS-PLS? is it Ref-CARS-PLS? clarify.  

The authors could have collected canopy spectra and used them in this study. Why the authors ended up using leaf spectra eventually?

It seems the authors collected data for four different growth stages and pooled them together to build their final models. Why not calibrate models for each growth stage and compare their performance based on the different growth stages?

Final selected features and corresponding spectral bands need to be discussed by comparing previous studies.

Discuss the generalizability of the study.

What are the implications of the results for large-scale and large-area remote sensing?

Round 2

Reviewer 1 Report

Descriptive text and data presentation are much better now. However, it seems the revision was done a bit hastly:

Reread the abstract (two times set now).

"spectrum measurements and leaves sampling" should read as spectral measurements and leaf sampling. (line 233 in my version)

In the end of the introduction, the objectives cannot read as "This paper is organized into four parts...". It needs to start with "the objectives of the present work ...." followed by the concrete objectives of the work.

Why is the caption of table 1 with capital first letters? The unit doesn't make sense here, since the sample number doesn't have this unit. Please place unit where appropriate.

line 505: here analysis should be with capital first letter in the revision

Please, integrate the description of subfigure "a) PLS" and "b)..." and "c9..." and "d)..." in the caption below the figure. Each figure needs to be readable as stand-alone with its caption, but without the surrounding text.

Reviewer 3 Report

The authors properly addressed my comments and clarified my raised issues. Only minor change that needs to be made on the title is eliminating "the", which is redundant. 

Author Response

Dear reviewer:
    Thank you for your suggestion again, which will further improve the quality of this paper.
    According to your suggestions, the redundant "the" on the title has been deleted.